# A New Albomycin-Producing Strain of *Streptomyces globisporus* subsp. *globisporus* May Provide Protection for Ants *Messor structor*

**DOI:** 10.3390/insects13111042

**Published:** 2022-11-11

**Authors:** Yuliya V. Zakalyukina, Nikolay A. Pavlov, Dmitrii A. Lukianov, Valeria I. Marina, Olga A. Belozerova, Vadim N. Tashlitsky, Elena B. Guglya, Ilya A. Osterman, Mikhail V. Biryukov

**Affiliations:** 1Center for Translational Medicine, Sirius University of Science and Technology, Olympic Avenue 1, 354340 Sochi, Russia; 2Department of Soil Science, Lomonosov Moscow State University, Leninskie Gory 1, 119991 Moscow, Russia; 3Center of Life Sciences, Skolkovo Institute of Science and Technology, Bolshoy Boulevard 30, Bld. 1, 121205 Moscow, Russia; 4Department of Chemistry, Lomonosov Moscow State University, Leninskie Gory 1, 119991 Moscow, Russia; 5Shemyakin-Ovchinnikov Institute of Bioorganic Chemistry of the Russian Academy of Sciences, Miklukho-Maklaya st. 16/10, 117997 Moscow, Russia; 6Institute of Translational Medicine, Pirogov Russian National Research Medical University, Ostrovityanova st. 1, 117997 Moscow, Russia; 7Department of Biology, Lomonosov Moscow State University, Leninskie Gory 1, 119991 Moscow, Russia

**Keywords:** actinobacteria, *Streptomyces globisporus* subsp. *globisporus*, albomycin, defensive symbiosis, ants, *Messor structor*

## Abstract

**Simple Summary:**

Insects are the most numerous and diverse animals on our planet. They have mastered different habitats and are able to resist many external threats. Many of them have long concluded mutually beneficial alliances with microorganisms that are capable of producing biologically active substances. We found that *Messor structor* ants have a very common actinobacteria that secretes albomycin, an antibiotic capable of suppressing the growth of enthomopathogenic bacteria in the smallest concentrations. Perhaps this is one of the secrets of ants’ resistance to external factors and their successful evolutionary development.

**Abstract:**

There are several well-studied examples of protective symbiosis between insect host and symbiotic actinobacteria, producing antimicrobial metabolites to inhibit host pathogens. These mutualistic relationships are best described for some wasps and leaf-cutting ants, while a huge variety of insect species still remain poorly explored. For the first time, we isolated actinobacteria from the harvester ant *Messor structor* and evaluated the isolates’ potential as antimicrobial producers. All isolates could be divided into two morphotypes of single and mycelial cells. We found that the most common mycelial morphotype was observed among soldiers and least common among larvae in the studied laboratory colony. The representative of this morphotype was identified as *Streptomyces globisporus* subsp. *globisporus* 4-3 by a polyphasic approach. It was established using a *E. coli* JW5503 pDualRep2 system that crude broths of mycelial isolates inhibited protein synthesis in reporter strains, but it did not disrupt the in vitro synthesis of proteins in cell-free extracts. An active compound was extracted, purified and identified as albomycin δ2. The pronounced ability of albomycin to inhibit the growth of entomopathogens suggests that *Streptomyces globisporus* subsp. *globisporus* may be involved in defensive symbiosis with the *Messor structor* ant against infections.

## 1. Introduction

The widespread use of antimicrobial compounds in medicine and agriculture has led to the emergence of multidrug-resistant pathogens, recognized now as a significant threat to human health [1,2]. The search for novel compounds possessing antimicrobial properties is still one of the ways we could overcome this global challenge, and microorganisms are the main source in this research [3,4,5]. Recently great attention was attracted by microbes that form symbioses with higher organisms, in particular, plants and animals [6]. The main reason for that is the mutual evolutionary path, wherein microbes have proved their usefulness to the host [7,8].

Some prokaryotes, principally the phylum Actinomycetota [9], are involved in the formation of so-called “defensive (or protective) symbioses” [10,11,12] with many eukaryotic organisms. The Insecta class, with the largest number of species, is remarkable among them for these interactions [13,14]. Through the release of various antibiotic compounds, actinobacteria protect insects, their brood and food substrate from potential pathogens and parasites [7,15,16,17,18]. Well known in this respect are leaf-cutting ants (Atta, Acromyrmex) of the subfamily Myrmicinae. Their existence, and in particular their feeding and development, depends entirely on the symbiotic actinobacteria of the genus *Pseudonocardia* [19], localized on the insect cuticle.

However, the specificity of “defensive symbioses” in other species of Formicidae remains unclear, including one of the dominant ants of the steppe zone, *Messor structor* (the steppe harvester ant). In this paper, we report on the isolation of actinobacteria from a laboratory colony of *Messor structor*, the identification of the produced antimicrobial compound (albomycin δ2), and its high activity against individual entomopathogens.

## 2. Materials and Methods

### 2.1. Ant Colony Rearing and Microbial Isolation

Prior to rearing an ant colony, a special incubator was designed; it consisted of a glass tube with sterile water and poppy seeds as the main nutrient substrate. The mated queen was maintained there during a 3-month period, until the adult ant quantity was sufficient to place the colony in a specific formicarium. The formicarium is made from an acrylic plastic with two main chambers—an arena, through which the seeds are supplemented, and a system of chambers, where ants raise their brood. In the center of the formicarium, a specific watering cell is present, which maintains the humidity and water level inside the nest. Humidity is maintained between 70 and 90% and temperature at 24 °C respectively, without direct sunlight on the formicarium.

The queen was collected during the mating season in July 2017 in the Astrakhan region, Russia (46°51′13.5″ N 47°59′06.2″ E). The isolation of actinobacteria strains was performed only after the stabilization of a population number of at least 50 specimens.

Actinobacterial strains were isolated from the bodies of larvae, pupae and imago worker and soldier castes of *Messor structor*. A total of 14 individuals from each group were examined. Every specimen was washed three times in sterile distilled water and then crushed by a tissue microhomogenizer with sterile saline solution. Aliquots of this mixture and their 10-fold dilutions were spread over mineral agar 1 [20] and organic agar 79 [21] supplemented nystatin and nalidixic acid at final concentrations 250 μg/mL and 10 μg/mL, accordingly, and incubated for 14 days at 28°C [22]. Actinobacteria isolates were purified and maintained on ISP 3 slants [23] and preserved as a suspension of mycelial fragments and spores in 20% glycerol at −20 °C.

### 2.2. 16S rRNA Phylogeny of Isolated Strains

The extraction of the genomic DNA of isolates and PCR amplification were achieved using procedures described elsewhere [24]. Both the pair of universal primers F27 (5′-AGAGTTTGATCMTGGCTCAG-3′) and R1492 (5′-TACGGYTACCTTGTTACGACTT-3′) and actinobacterial primers 243F (5′-GGATGAGCCCGCGGCCTA-3′) and A3R (5′-CCAGCCCCACCTTCGAC-3′) were used. The amplicons were purified and sequenced using a commercial service (EvroGen). All sequences were identified by searching close relatives with the BLAST service (https://blast.ncbi.nlm.nih.gov/Blast.cgi, accessed on 1 October 2022) and were submitted in GenBank with the assignment of access numbers.

### 2.3. Genome Features and Phylogenomic Analysis

Genome of strain 4-3 was sequenced de novo by Skoltech Genomics Core Facility, using the Illumina HiSeq 4000 platform (Illumina, San Diego, CA, USA). The quality control and adapter trimming was done by the bbDuk tool from the BBMap suite v38.42 (https://sourceforge.net/projects/bbmap/, accessed on 1 February 2022). Genome assembly was performed by SPAdes v3.13.0 [25]. The genome was annotated using RASTtk pipeline implemented on the PATRIC web service [26]. Assembly is available in the European Nucleotide Archive with project accession PRJEB51905.

Values of average nucleotide identity (ANI), genome completeness and quality were evaluated using a web service MiGA (http://microbial-genomes.org/, accessed on 1 October 2022), and in silico digital DNA:DNA hybridization (DDH) values were calculated by using the GGDC method, with the recommended formula 2, available at the TYGS web service (https://tygs.dsmz.de/, accessed on 1 October 2022) [27].

Phylogenomic analysis was performed using Type (Strain) Genome Server (https://tygs.dsmz.de/, accessed on 1 October 2022). The phylogenomic tree inferred with FastME 2.1.6.1 [28] from GBDP distances calculated from genome sequences. The branch lengths are scaled in terms of GBDP distance formula d5.

The full-length 16S rRNA gene sequences of strain 4-3 was extracted from the whole genome sequence (PRJEB51905) and was compared to sequences of related *Streptomyces globisporus* subsp. *globisporus* and some actinobacteria species, firstly isolated from insects. Evolutionary trees based on 16S rRNA gene sequences were inferred with the neighbor-joining [29], maximum-parsimony [30] and maximum-likelihood [31] treemaking algorithms after CLUSTAL W alignment by using MEGA software version X [32].

### 2.4. Analysis of Bioactive Compound Biosynthetic Gene Clusters

Secondary metabolite biosynthetic gene clusters in complete genome strain 4-3 and its neighbors were identified with the bacterial version of antiSMASH 6.1.0 (https://antismash.secondarymetabolites.org/, accessed on 1 October 2022). Homologous regions on each genome were identified using NCBI Blastn (https://blast.ncbi.nlm.nih.gov/, accessed on 1 October 2022).

### 2.5. Phenotypic Characterization

Cultural characteristics of strain 4-3 were observed in ISP 2–ISP 7 media [23] after cultivation for up to 14 days at 28 °C. The RAL Classic Standard was used to determine the designations of colony colors. The shape of spore chains and the spore surface of strain 4-3 on ISP 3 after cultivation at 28 °C for 14 days were studied using light microscopy (Fisherbrand AX-502, Fisher Scientific, Merelbeke, Belgium) and scanning electron microscopy (JSM-6380LA, JEOL, Tokyo, Japan).

Carbon source utilization was assessed on basal medium ISP 9 [23] with the addition of 0.04% solution of bromocresol purple at 28 °C for 14 days. Enzyme activities were estimated using a paper indicator system (NPO Microgen, Nizhny Novgorod, Russia) according to the manufacturer’s recommendations at 28 °C for 7 days. The degradation of casein, starch and cellulose was estimated on the clearing of the insoluble compounds around areas of growth [33].

### 2.6. Biological Activity Testing

#### 2.6.1. Screening of the Antimicrobial Potential

The ability of actinobacteria isolates to inhibit bacterial growth was assessed by the agar diffusion method. The isolates were challenged against different clinically significant microorganisms: *Staphylococcus aureus* ATCC 25923, *Bacillus subtilis* ATCC 6633, *Candida albicans* CBS 8836 and *Aspergillus niger* INA 00760. Anti-entomopathogenic activity was also investigated against: *Bacillus thuringiensis* VKM B-6650, *Paenibacillus alvei* VKM B-502, *Beauveria bassiana* VKM F-1357 and *Entomophthora coronata* VKM F-1359.

Test bacteria, yeast and fungi strains were individually inoculated in Luria–Bertani agar, dextrose–peptone-yeast agar and potato dextrose agar, accordingly. Agar plugs of each actinobacterial isolate (2-week-old cultures in ISP6) were placed on the surface of the inoculated media. The plates were incubated at 37 °C, and after 24–48 h, the inhibition zones were checked.

#### 2.6.2. Reporter Assays on Agar Plates

The two *E. coli* reporter strains: BW25113 wild-type pDualRep2 and JW5503 ΔtolC pDualRep2 [34] were used in this work as previously described [35]. Briefly, 100 µL of cultural broth were placed into wells in agar that had the lawn of a reporter strain. Two control antibiotics, erythromycin (Ery, 2 μg) and levofloxacin (Lev, 0.05 μg), were additionally applied to an agar plate. Plates were incubated at 37 °C overnight and then scanned by ChemiDoc (Bio-Rad) in the modes ‘Cy3-blot’ for RFP and ‘Cy5-blot’ for Katushka2S. The expression of the rfp gene occurred in the case of the activation of the SOS-response system of the cell and katushka2S, in the case of a violation of translation, when the ribosome was stalled on the mRNA template. When scanning, the signal from two black and white images was superimposed on each other, with the assignment of green for the signal from the RFP protein and red for Katushka2S.

#### 2.6.3. Determination of Minimal Inhibitory Concentration (MIC)

Overnight cultures of *E. coli* ΔtolC, *Bacillus thuringiensis* and *Paenibacillus alvei* were diluted 1:1000 in LB medium. A sterile 96-well plate was then loaded with 200 mL of the diluted cultural media, with the initial row having 400 mL prior serial dilution. A stock solution of a HPLC-purified 4-3 sample was variously seeded in the initial row, along with erythromycin (Ery), which was used as a control for the experiment. Other wells were left without antibiotic but with diluted LB-culture media, while the rest were left with LB media only as additional controls. A two-fold serial dilution was then carried out, with gentle mixing in each row. The plates were then incubated overnight at 37 °C with shaking at 200 rpm. Cell growth was measured at 590 nm using a microplate reader (VICTOR X5 Light Plate Reader, PerkinElmer, Waltham, MA, USA). UV absorption of 4-3 was measured with a spectrophotometer (NanoPhotometer™ NP80, Implen, München, Germany), and the concentration was calculated using the known coefficients of extinction at 306 and 425 nm.

#### 2.6.4. Cell-free Translation

Cell-free (in vitro) translation reactions were performed in the presence of HPLC 4-3 fraction (1/10 of final volume) in 5 μL using the PURExpress® In Vitro system (NEB, Ipswich, MA, USA) supplemented with 100 ng Fluc mRNA and 0.05 mM D-luciferin. Chemiluminescence was recorded with VICTOR X5 Light Plate Reader. The Fluc mRNA obtained byMEGAscript™ T7 Transcription Kit (ThermoFisher, Carlsbad, CA, USA) from the circular DNA template.

### 2.7. Purification and Identification of Albomycin

To obtain a sufficient amount of the active compound for detailed bioactivity studies, strain 4-3 was cultured in four 750 mL Erlenmeyer flasks with 250 mL of liquid ISP 6 at 28 °C for 14 days under static conditions. One liter of fermentation broth 4-3 was subjected to gravity-flow reverse-phase chromatography on the sorbent LPS500H (polyvinylbenzene, pore size 50–1000 Å) (LLC “Technosorbent”, Moscow, Russia). The fermentation broth was applied to chromatographic media, which then was eluted consistently with 10, 20, 30, 40, 50, 75 and 100% solutions of acetonitrile in water. The pDualrep2 double reporter system was used to analyze the activity of the collected fractions. The most active fractions were eluted by 10 and 20% acetonitrile, they induced the expression of reporter protein Katushka2S. The 10% acetonitrile fraction was further purified by high-performance liquid chromatography (HPLC) (Agilent 1260, isocratic elution 4% of MeCN, 10 mM AcONH4, 1 mL/min, 25 °C) using a Phenomenex HPLC column (Luna 5 μm C18 (2) 100 Å, 4.6 × 250 mm), and the collected fractions were analyzed using the reporter pDualrep2.

Fractions with antibacterial activity corresponding to an individual peak on chromatograms were collected, and the active compound was identified using ultra-high-performance liquid chromatography–electrospray ionization–high-resolution mass spectrometry (UPLC–ESI–HRMS). Analysis was carried out on an Ultimate 3000 RSLCnano HPLC system connected to an Orbitrap Fusion Lumos mass spectrometer (ThermoFisher Scientific). A sample of the active compound was separated on Luna Omega C18 100 × 2.1 mm 1.6 μm columns at a 0.2 mL/min flow rate and at RT. Separation was done by a gradient elution in a two-component mixture from the initial 5% to 20% of component B for 10 min. Component A was 0.1% formic acid plus 10 mM formate ammonium in water, and component B was a mixture of 0.1% formic acid in 100% MeCN and 10% 10 mM formate ammonium in water in ratio 9:1. UV data were registered at 290 nm. MS1 and MS2 spectra were collected in positive ion mode and recorded at 30 K and 15 K resolution, respectively, with HCD fragmentation.

## 3. Results

### 3.1. Isolation of Actinobacteria Strains, Associated with Messor Structor Ants

All bacterial strains isolated from *Messor structor* individuals were divided into two morphotypes: one of which formed beige branching mycelium, dark-pigmented and straight spore chains (later labeled as 4-3), and the other, formed rough colorless colonies from Gram-positive cocci (L1). Results demonstrated that strains of 4-3-morphotype had more association with the caste of soldiers with 89% frequency among this group, while workers’ caste, pupae and larvae showed less specific results—50%, 21%, and 7% respectively. On the contrary, L1-bacteria were found in the vast majority of individuals from these groups besides soldiers (Appendix A).

A comparison with the GenBank database demonstrated that all strains assigned to the 4-3-morphotype, in addition to their phenotypical similarity, had an identical sequence of the 16S rRNA genes, indicating their belonging to the *Streptomyces* genus. The closest strains were *Streptomyces globisporus* subsp. *globisporus* DSM 40136 (formerly a type strain of *Streptomyces albovinaceus*), *Streptomyces globisporus* subsp. *globisporus* DSM 40199^T^, *Streptomyces rubiginosohelvolus* DSM 40176^T^ and *Streptomyces pluricolorescens* DSM 40019^T^. The partial 16S rRNA sequences of L1-morphotype strains showed 100% similarity with *Staphylococcus gallinarum* DSM 20610^T^. Among the representatives of the mycelial morphotype, strain 4-3 was selected for a more detailed study of the genome features and antagonistic activity.

### 3.2. Genome Features and Phylogenomic Analysis of Streptomyces sp. Strain 4-3

Phylogenomic analysis based on whole-genome sequences showed that strain 4-3 formed a well-supported monophyletic clade with *S. globisporus* subsp. *globisporus* DSM 40199^T^ and *S. globisporus* subsp. *globisporus* DSM 40136 with 95% bootstrap value (Figure 1).

The complete genome size of strain 4-3 was 7,941,828 bp with DNA G + C content of 71.6%, which was consistent with the G + C content of the genus *Streptomyces* [36]. The closest neighbors *S. globisporus* subp. *globisporus* DSM 40199^T^ and *S. globisporus* subp. *globisporus* DSM 40136 are characterized by similar genome size and G + C content (Appendix A).

The ANI and in silico dDDH values between strains 4-3 and *S. globisporus* subp. *globisporus* DSM 40136 and *S. globisporus* subp. *globisporus* DSM 40199^T^ were above the recommended threshold of 96% and 70% (Table 1) needed for species separation [27,37]. Based on this, strain 4-3 most likely belongs to the species *Streptomyces globisporus* subp. *globisporus*.

Neighbor-joining phylogenetic analysis demonstrated that 4-3 was most closely related to type and not type strains of *Streptomyces globisporus* subsp. *globisporus*: DSM 40199^T^, DSM 40139, C-1027, TFH56, as well as to *S. rubiginosohelvolus* DSM 40176^T^, *S. pluricolorescens* DSM 40019^T^, *S. sindenensis* DSM 40255^T^, *S. anulatus* DSM 40361^T^ and *S. griseus* subsp. *griseus* ATCC 13273 and formed a share clade with 100% bootstrap value (Appendix A). However, type strains of actinobacterial species, first isolated from ants and other insects, did not form well-supported clades with 4-3. This relationship was also supported in the phylogenetic trees generated with maximum-parsimony and maximum-likelihood methods (Appendix A).

### 3.3. Phenotypic Characterization of Streptomyces sp. Strain 4-3

To further evaluate the features of the 4-3 strain using a polyphasic taxonomy approach, the cultural, morphological and physiological properties of 4-3 were compared with ones of the type strains of *Streptomyces globisporus* subsp. *globisporus*. Results demonstrated the identity of these organisms in morphology—the shape of sporophores and spore surface (Appendix A, Figure 2) and the high similarity of their cultural characteristics on the series of ISP media (Table 2, Appendix A).

However, some differences should be noted in the biochemical and physiological properties of strain 4-3 and the closest type strains of *S. globisporus* subsp. *globisporus* (Appendix A). For example, the production acid from glucose and xylose was positive in strain 4-3 (Appendix A), whereas the other type strains showed negative results. In the decomposition of polymers, strain 4-3 was unable to use cellulose as a sole carbon source, whereas the other type strains utilized it. Furthermore, the enzyme assay of 4-3 was negative for L-ornithine decarboxylase and L-arginine decarboxylase; in contrast, DSM 40199^T^ and DSM 40136 demonstrated positive responses (Appendix A). However, most of the biochemical tests showed similar results.

According to the obtained data, we may conclude that strain 4-3 isolated from *Messor structor* ants can be classified as *Streptomyces globisporus* subsp. *globisporus*.

### 3.4. Analysis of 4-3 Bioactive Compound Biosynthetic Gene Clusters

The bioinformatics analysis of *Streptomyces globisporus* subsp. *globisporus* 4-3 genome revealed a biosynthetic gene cluster of albomycins, consisting of 18 genes from *abm*A to *abm*R, completely identical to that of *Streptomyces* sp. ATCC 700974 (Figure 3), described in detail earlier [38]. The presence of *abm*K, participating directly in the formation of SB-217452 (the active seryl-tRNA synthetase inhibitor component of albomycin) [39] and also providing self-resistance to albomycins [40], indicates the ability of *Streptomyces globisporus* subsp. *globisporus* 4-3 to actively produce albomycin δ2.

Furthermore, the 4-3 strain genome contains a number of second metabolic biosynthesis gene clusters (SMBGCs), coded production of antimicrobial compounds (streptophenazines B/C/E/H/G, mayamycins), odor substances (geosmin), pigments (melanin, isorenieratene), siderophores (streptobactin, coelichelin), cytoprotectants (ectoine) and others (Appendix A).

### 3.5. Screening Antimicrobial Activity

All isolated strains were initially tested for the ability to secrete antimicrobial substances. It was noticed that all phenotypically similar mycelial strains showed the same activity pattern, so we focused on the study of strain 4-3. Analysis of antimicrobial activity demonstrated that the 4-3 strain noticeably inhibited the growth of various pathogenic microorganisms (Appendix A): bacteria (*Bacillus subtilis*, *Staphylococcus aureus*) and fungi (*Aspergillus niger*), but it is especially active on entomopathogenic microorganisms (*Bacillus thuringiensis*, *Paenibacillus alvei*, *Beauveria bassiana*, *Entomophthora coronata*). To evaluate the MIC of the 4-3 compound, we chose strains that were the most susceptible during the screening procedure. The estimated concentration of the HPLC-purified sample was 0.6 μg/mL. In the prepared series of microdilutions, the test strains demonstrated high sensitivity to the 4-3 compound (Appendix A), including entomopathogenic bacteria.

The agar plugs and cultural broth aliquots of 4-3 demonstrated prominent antibiotic activity in tests on the reporter strains (Figure 4) and exhibited strong Katushka2S reporter induction, indicating that the active compound produced by the isolate functions as an inhibitor of protein biosynthesis. To further evaluate the possible mechanism of action, the in vitro translation analysis was performed.

### 3.6. Analysis of Bioactive Compounds: Cell-free Translation

It was decided to test the HPLC-purified sample 4-3 in in vitro translation procedure to completely evaluate its translation inhibitory activity. Despite induction of the Katushka2S, indicating the inhibition of protein synthesis in the cells of the reporter strains (Figure 4), the active compound of the strain 4-3 did not suppress translation in the cell-free system (Figure 5).

### 3.7. Purification and Identification of Albomycin

The liquid culture of *Streptomyces globisporus* subsp. *globisporus* 4-3 was preconcentrated and purified by solid-phase extraction (SPE) on LPS500H resin. It was binded on LPS500H, washed with water and the active compound was eluted with 10% MeCN solution in water. Further purification of active SPE fraction was carried out by RP HPLC on a C18 column in isocratic mode with aqueous solution of ammonium acetate—MeCN as eluent. Thus, a pure compound was isolated with a UV maxima at 283 and 424 nm, and the activity testing confirmed that it corresponded to an active metabolite (Appendix A).

The metabolite identification was carried out using LC-HRMS/MS analysis (Appendix A). The observed exact masses 1046.3102 of molecular ion [M + H]^+^ of the compound and characteristic isotope distribution corresponded to the composition C_37_H_57_FeN_12_O_18_S (calculated exact mass 1046.3057). The main fragmentary ion in the MS2 spectrum due to the loss of the cytosine part was observed at *m*/*z* 878. MS1–MS2 raw data were analyzed in Compound Discoverer 3.2 software (Thermo Fisher Scientific). Peak annotation was performed with ChemSpider, Natural Product Atlas 2020 and COCONUT databases using the mass spectra information with 5 ppm mass accuracy, isotopic distribution ≥ 50% and match score ≥ 85%. The result of the analysis allowed us to conclude that the active compound is a known inhibitor of bacterial seryl-tRNA synthetase, albomycin δ2 (Figure 6) [41].

## 4. Discussion

Albomycin δ2 belongs to the group of sideromycins, antibiotics covalently bound to siderophore fragments and penetrating into the cell through siderophore absorption pathways, implementing the so-called “Trojan horse” strategy [42].

Albomycin, originally reported as grisein, were first isolated from soil from the soil-dwelling *Streptomyces griseus* in 1947 by S. Waksman and colleagues. It was also identified in *Streptomyces subtropicus* (previously known as *Actinomyces subtropicus*) by Gause and Brazhnikova in 1951, and their identity was confirmed later [43]. It is noteworthy that albomycin was also known as alveomycin, antibiotics A 1787, LA 5352 and LA 5937, and Ro 5-2667 in the literature [44].

Albomycins have attracted significant attention due to their potent antibacterial activities against both Gram-negative and Gram-positive bacteria, including multi-drug-resistant strains [45]. Moreover, no toxicity was observed during in vivo studies of albomycins, and it was well tolerated and safe up to a maximum dose evaluated in mice [41].

The structure of albomycin and related compounds (δ1, δ2, and ε) was fully established more than 30 years after the initial discovery [46]. Albomycins have a thioribosyl nucleoside moiety linked to an iron-chelating ferrichrome-type siderophore through a serine residue. The L-serine-thioheptose dipeptide partial structure, known as SB-217452, has been found to be the active seryl-tRNA synthetase inhibitor [39].

The iron-chelator portion serves as a vehicle for the active delivery of the albomycin warhead inside both Gram-positive and Gram-negative bacterial cells through the ferrichrome-specific transporter system. The formation of the active inhibitory form of albomycin, SB-217452, occurs intracellularly under the action of PepN peptidase (*E. coli*), which cleaves off the siderophore part. As a result, the toxic nucleoside part is accumulated in the cytoplasm of *E. coli* in ~500-fold excess over the concentration of the antibiotic in the medium [42]. This explains the fact that albomycin does not act on translation in a cell-free extract—in such a system, there are no enzymes that cleave off the siderophore (Figure 4). Likewise, added directly to bacterial culture, the nucleoside portion of albomycin does not inhibit cell growth [42] since it cannot get inside the cells without the siderophore part.

In the crops of *Messor structor* ants contained in laboratory conditions, we constantly observed mycelial bacteria of a certain phenotype, most closely associated with individuals from the soldier caste. A representative of this phenotype, strain 4-3, was identified using a polyphase approach as *Streptomyces globisporus* subsp. *globisporus*.

The members of *Streptomyces* are widely known for their ability to produce various antibiotic compounds and often in association with insects: ants, wasps, beetles [7]. According to the literature, *Streptomyces globisporus* was most often isolated from soils, plants and so on [47] but rarely from insects. There is an example of the *Streptomyces globisporus* SP6C4, which plays a significant role in the mutualistic relationship between pineapple strawberries (*Fragaria ananassa*) and bees (Apidae), protecting both the plant and the insect from pathogenic microorganisms, including the phytopathogenic fungus *Botrytis cinerea* [48]. In addition, it is reported about *Streptomyces globisporus* WA5-2-37, isolated from the intestinal tract of the American cockroach (*Periplaneta americana*), produced actinomycin X2 and collismycin A, which showed great activity against MRSA ATCC 43300 [49]. This is the first reported naturally occurring strain of *S. globisporus* subsp. *globisporus* isolated from Formicidae.

The genus *Messor* (Forel, 1890) is a moderately large genus, with more than 126 species worldwide recognized; they are mainly distributed in the Palearctic, Afrotropical and Oriental regions. Messor species are granivorous and play an important role in ecosystem maintenance and plant-seed dispersal. The harvesting ant, *Messor structor* (Latreille, 1798), is an ecosystem engineer in many dry biocenoses [50].

The main food resource for harvester ants are grains of cereals and oilseed plants. Such seeds have a solid endosperm and require considerable effort to grind them. Representatives of the soldier caste have a large head—the result of the development of massive occipital muscles responsible for the work of the lower jaw—and powerful mandibles. They initially grind the seeds, and then smaller worker ants process the prepared pieces of seeds, since it requires less effort; turn them into flour; moisten them with saliva and uses them as food for the colony. Their saliva is dominated by amylase enzymes that break down starch [51].

The greatest abundance of actinobacteria, isolated from soldiers and workers, which represent a conveyor for the production of food for the colony, may indicate a possible symbiotic relationship between ants and streptomycetes. Actinobacteria receive food and shelter, and in return produce a substance with a wide spectrum of action that protects the food resource of ants from spoilage. The ability of associated streptomycetes to synthesize albomycin can be extremely useful for hosts—since harvester ants contact with soils and plants and encounter a large number of microorganisms, Gram-positive and Gram-negative, as well as fungi.

As is known, albomycin δ2 is characterized by surprisingly low inhibitory concentrations for many pathogenic microorganisms: minimum inhibitory concentrations (MICs) as low as 5 ng/mL against *Escherichia coli* and 10 ng/mL against *Streptococcus pneumoniae* [52]. In our in vitro experiments, *Streptomyces globisporus* subsp. *globisporus* 4-3 very actively suppressed the growth of various entomopathogens: *Paenibacillus alvei* VKM B-502, *Bacillus thuringiensis* VKM B-6650, *Beauveria bassiana* VKM F-1357 and *Entomophthora coronata* VKM F-1359 (Appendix A), which can provide one of the forms of protection of the ant family health.

A culture-dependent study of the actinobiome of a *Messor structor* colony, living under laboratory conditions, revealed the strains dominant in adult individuals from soldier and worker casts. All mycelial isolates demonstrated the same genotypic and phenotypic properties and were identified as *Streptomyces globisporus* subsp. *globisporus*. They produced protein synthesis inhibitor albomycin δ2, which was active against entomopathogens. The distribution of these actinobacteria among individuals from different castes suggests their essential role in maintaining the health of the ant family. The confirmation of this assumption, as well as the hypothesis about the methods of the transmission of strains between individuals, needs more detailed studies on a wider range of colonies of harvester ants living in formicaria.

## Figures and Tables

**Figure 1 insects-13-01042-f001:**
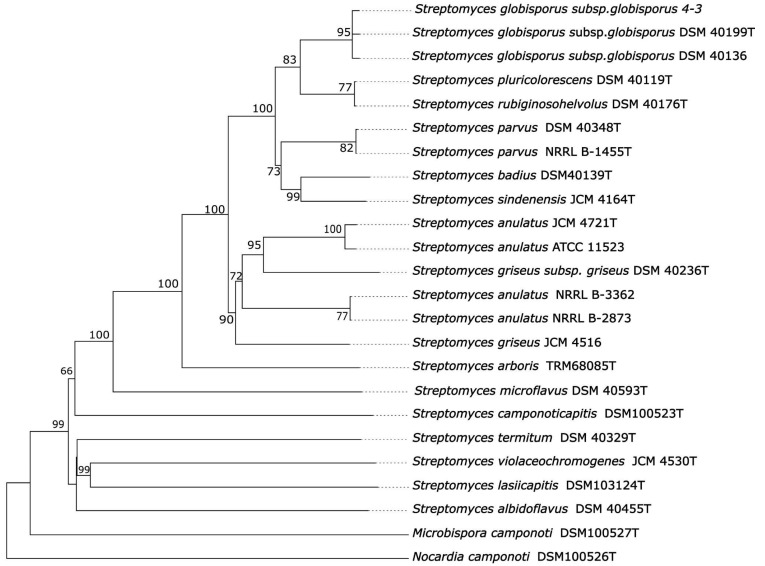
Phylogenetic tree based on whole-genome sequences from 4-3, related type strains and actinobacteria isolated from ants. The branch lengths are scaled in terms of GBDP distance formula d5. Numbers above branches are GBDP pseudo-bootstrap support values > 60% from 100 replications, with an average branch support of 81.4%. *Nocardia camponoti* DSM 100526^T^ as outgroup [27].

**Figure 2 insects-13-01042-f002:**
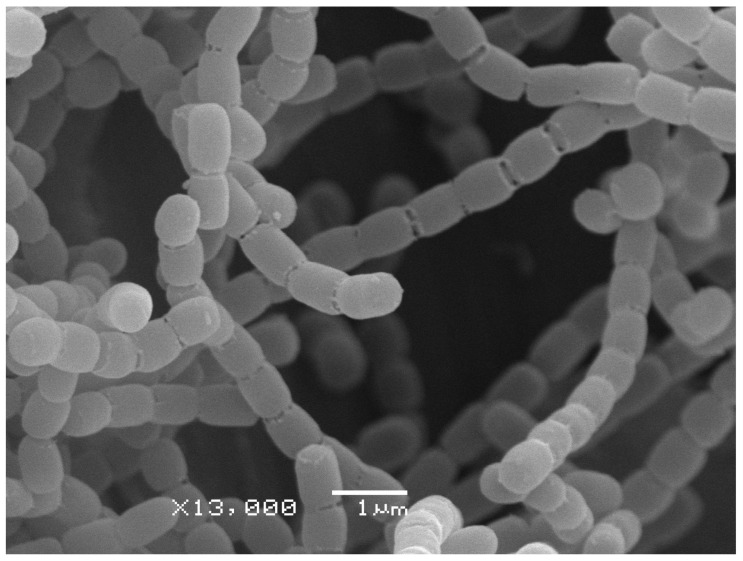
Scanning electron micrograph of the strain *Streptomyces globisporus* subsp. *globisporus* 4-3, showing the spore surface after incubation on ISP 3 medium at 28 °C for 14 days.

**Figure 3 insects-13-01042-f003:**
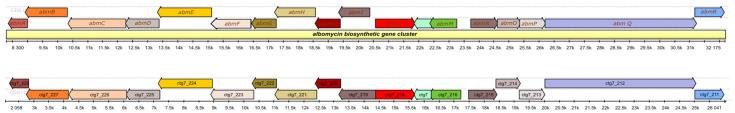
Biosynthetic gene clusters of albomycins: genetic organization of the albomycin (abm) gene cluster in *Streptomyces* sp. ATCC 700974 (**top**) and gene cluster in *Streptomyces globisporus* subsp. *globisporus* 4-3 (**bottom**). The homologous abm and ctg genes are filled with the same colors.

**Figure 4 insects-13-01042-f004:**
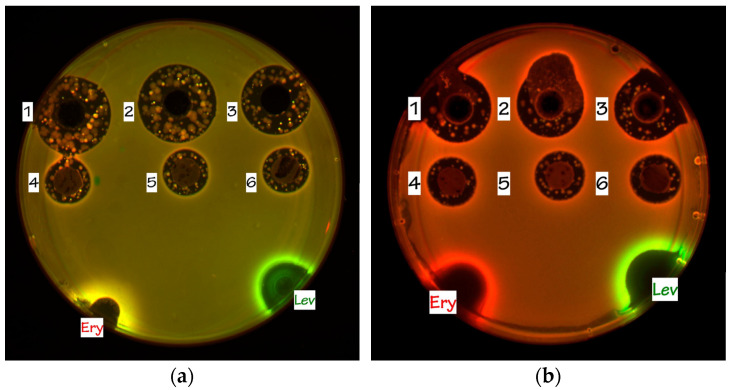
In Vitro testing of ant-associated strains activity using: (**a**) *E. coli* BW25113 wild-type pDualRep2 reporter strain; (**b**) *E. coli* JW5503 ΔtolC pDualRep2 reporter strain. 1, 2, 3—crude broth aliquots and 4, 5, 6—agar plugs of X2, X1 and 4-3, accordingly. The agar plates were spotted with erythromycin, 5 μg/mL (Ery) and levofloxacin, 2 μg/mL (Lev).

**Figure 5 insects-13-01042-f005:**
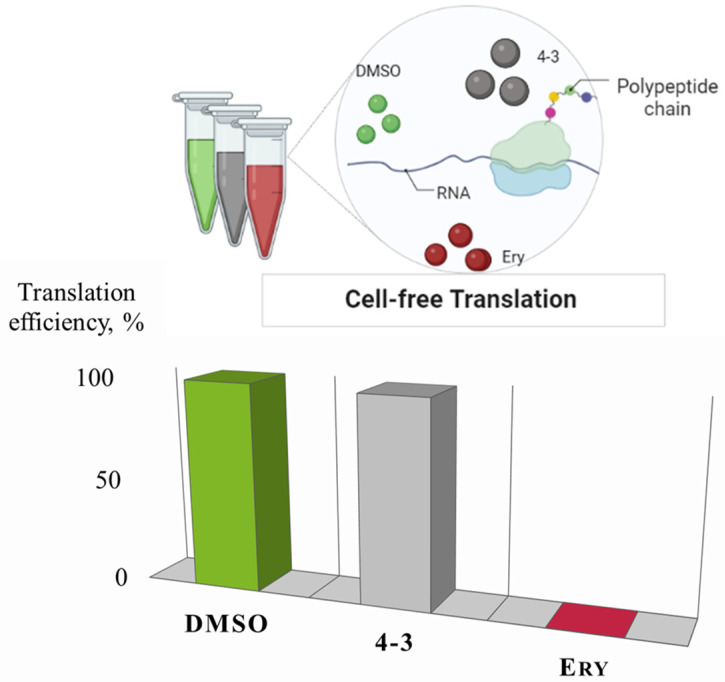
General scheme of cell-free translation assay (**top**) and translation effectiveness (**bottom**) in the presence of sample 4-3, DMSO 1% (negative control) and erythromycin (Ery, 5 μg/mL) as positive control.

**Figure 6 insects-13-01042-f006:**
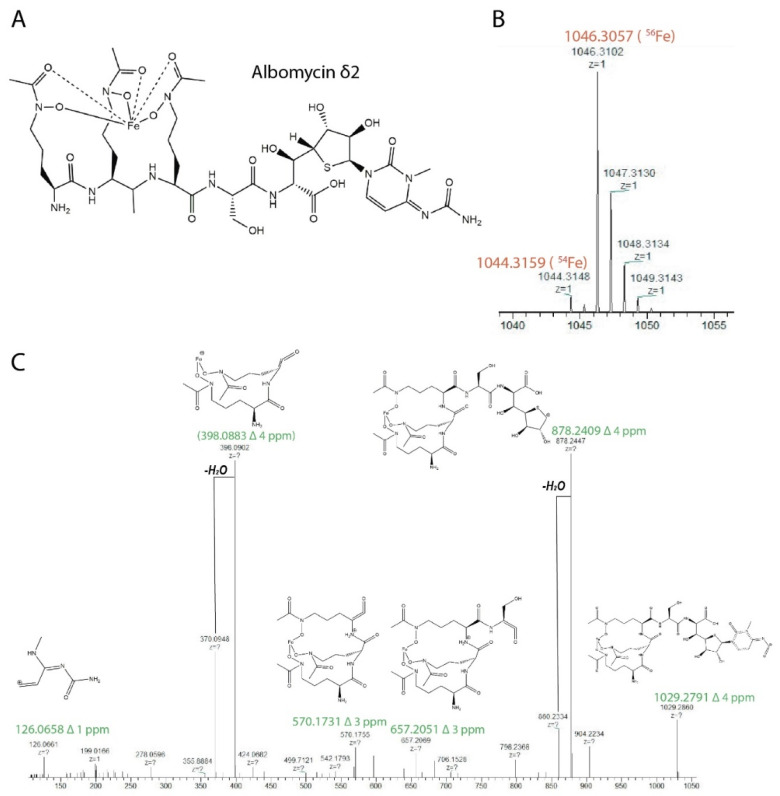
Identification of albomycin δ2. (**A**) Molecular structure of albomycin δ2. (**B**) Albomycin δ2 molecular ion [M + H]^+^ observed at 1046.3102 and its isotopic distribution (calculated mass are signed as red). (**C**) The positive-mode HCD mass spectra of the parent ion at *m*/*z* 1046.3102 with the molecular fragment structures (calculated mass for the fragments are signed as green).

**Table 1 insects-13-01042-t001:** Genome relatedness of 4-3 and *Streptomyces* type-strains.

Subject Strain	ANI,%	dDDH (in %)	G + C Content Difference (in %)
*Streptomyces globisporus* subsp. *globisporus* DSM 40136	99.3	96.3	0.14
*Streptomyces globisporus* subsp. *globisporus* DSM 40199^T^	99.4	95.9	0.12
*Streptomyces rubiginosohelvolus* DSM 40176^T^	96.3	66.2	0.18
*Streptomyces pluricolorescens* DSM 40019^T^	96.2	66.1	0.14
*Streptomyces parvus* NRRL B-1455^T^	94.8	56.2	0.01
*Streptomyces parvus* JCM 4069^T^	94.7	55.7	0.05
*Streptomyces sindenensis* JCM 4164^T^	94.2	52.9	0.25
*Streptomyces badius* JCM 4350^T^	94.2	52.5	0.11
*Streptomyces anulatus* JCM 4721^T^	92.1	41.8	0.15

**Table 2 insects-13-01042-t002:** Cultural characteristics of strain 4-3 (1) and closely related *Streptomyces globisporus* subsp. *globisporus* DSM 40199^T^ (2) and DSM 40136 (3).

Media	1	2	3
Yeast extract-malt extract (ISP 2)			
Growth	good	good	good
Aerial spore-mass color	oyster white	oster white	cream
Substrate mycelial color	beige	ochre yellow	beige
Soluble pigment	none	none	none
Oatmeal (ISP 3)			
Growth	good	good	good
Aerial spore-mass color	oyster white	oster white	cream
Substrate mycelial color	brown beige	ivory	beige
Soluble pigment	brown beige	none	none
Inorganic salts-starch (ISP 4)			
Growth	good	good	good
Aerial spore-mass color	white	light gray	sparse
Substrate mycelial color	colorless	green brown	beige
Soluble pigment	none	none	none
Glycerol-asparagine (ISP 5)			
Growth	good	good	good
Aerial spore-mass color	light olive	none	none
Substrate mycelial color	sand yellow	ivory	beige
Soluble pigment	sand yellow	none	none
Peptone-yeast extract iron (ISP 6)			
Growth	good	good	good
Aerial spore-mass color	white	none	none
Substrate mycelial color	beige	sand yellow	beige
Soluble pigment	none	none	none
Tyrosine (ISP 7)			
Growth	weak	good	good
Aerial spore-mass color	ivory	none	cream
Substrate mycelial color	yellow–red	beige	beige
Soluble pigment	none	none	none

Data for *Streptomyces globisporus* subsp. *globisporus* DSM 40199^T^ and DSM 40136 are from DSMZ catalogue (https://www.dsmz.de/collection/catalogue/microorganisms/catalogue, accessed 1 October 2022).

## Data Availability

Genomic data of *Streptomyces globisporus* subsp. *globisporus* 4-3 can be found at https://www.ebi.ac.uk/ena/browser/home (accessed on 1 November 2022) with project accession PRJEB51905.

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
