# Peer review of "A New Albomycin-Producing Strain of Streptomyces globisporus subsp. globisporus May Provide Protection for Ants Messor structor"

_insects, 2022, doi:10.3390/insects13111042_

Round 1
Reviewer 1 Report
The article has a scientific value and novelty, is written legibly and all information are presented in a logical way. I have read the article with interest and pleasure. The introduction describes an actual state of the art in this field, some statements should be covered with proper publications. Methodology is good but some explanation should be provided.
Line 82 “Actinobacterial strains were isolated from bodies of larvae, pupae…” and in line 205 we can read “The analysis of 16S rRNA sequences and comparison them with GenBank database demonstrated that 4-3 strain belong to Streptomyces genus”. What about other strains? Did the researchers isolate only strains from Streptomyces genus? Some actinobacteria are connected with the healthy insect microbiota as for example Bifidobacterium (Actinobacteria) and others with some intestinal dysfunctions, please discuss your data with https://www.mdpi.com/2076-0817/10/3/381.
The Discussion should be extended with one paragraph of general conclusions.
Author Response
Response to Reviewer 1:
Thank you for your review of our paper. We have answered each of your points below.
Line 82 “Actinobacterial strains were isolated from bodies of larvae, pupae…” and in line 205 we can read “The analysis of 16S rRNA sequences and comparison them with GenBank database demonstrated that 4-3 strain belong to Streptomyces genus”. What about other strains?
The subsection 3.1. was amended to clarify:
- only two types of colonies were found on the used media: mycelial and unicellular with similar phenotypic features.
- initially, the number of strains was large, but then, based on the similarity of phenotypic characteristics and nucleotide sequences, they were combined and designated as 4-3 and L1-morphotypes. Later, one of the representatives of the mycelial morphotype, strain 4-3, underwent a more detailed study of the genome and antagonistic activity.
Did the researchers isolate only strains from Streptomyces genus?
Indeed, first of all we focused on the isolation of mycelial prokaryotes (for which we used medium with starch), but also used a rich nutrient medium (with glucose and peptone) suitable for a wide range of gram-positive bacteria. But on both media we observed representatives of the same mycelial morphotype (Streptomyces globisporus subsp. globisporus). On the medium of organ 79 there were also single-celled morphotype Streptococcus gallinarum.
Some actinobacteria are connected with the healthy insect microbiota as for example Bifidobacterium (Actinobacteria) and others with some intestinal dysfunctions, please discuss your data with https://www.mdpi.com/2076-0817/10/3/381.
Thank you for the link to an interesting study, we have studied it carefully. It is extremely significant, but uses a different methodology in relation to other host organisms. The use of cultural-independent methods of studying microbiomes provides information about the main players, and in this work we can see changes in the dominants under various conditions.
However, mycelial actinomycetes are often not dominant in microbial communities, but at the same time they play an important regulatory role due to the isolation of secondary metabolites.
We deliberately limited ourselves to traditional microbiological methods of isolation of pure cultures within the framework of this study, however, in the future. we plan to investigate the metagenome of harvester ants.
- The Discussion should be extended with one paragraph of general conclusions.
General conclusions have been added to the Discussion section (Lines 481-490).
Reviewer 2 Report
This is a carefully written paper that identifies the presence of an endosymbiont bacteria (Streptomyces globisporus subsp. globisporus) in a harvester ant species. The results presented here will be of great interest for researchers working in the fields of myrmecology, microbiology and ant-microbe associations. They show that the Streptomyces globisporus strain found in Messor structor ants synthesizes the active compound albomycine d2, which inhibit the growth of entomopathogens.
The manuscript is well written, clearly and concisely presented. The results are well explained and discussed. My major concern with this study is that all results and conclusions are based on the analysis on only one ant colony. I wonder why they did not collect more than one queen in the survey. I suggest the authors at least mention this as a week point of the study and the need of replicate these analyses in more colonies/locations.
I am not an expert on the microbiology and biochemistry methods employed in this paper so I dont feel qualified to judge them. I hope that at least one other reviewer has adequate experience in those methods to offer constructive criticism. I hope
Some minor comments:
L. 53: I suggest replacing “is” with “are” and delete “taxon of”
L. 60: Replace the first “the” with “a” and delete “ant”. Delete “of”
L. 60-65: This paragraph contains information that should go in the abtract/results/discussion sections, not in the introduction.
L. 92: should be: The extraction of genomic DNA of isolates “and” PCR amplification were achieved…
L. 205: delete “them”
L. 365: replace “has” with “have”
L. 398-399: Should be: The genus Messor Forel, 1890 is a moderately large genus, “with” more than 126 species recognized, …
Author Response
Response to Reviewer 2:
Thank you for your comments. Our answers to your points are as follows.
My major concern with this study is that all results and conclusions are based on the analysis on only one ant colony. I wonder why they did not collect more than one queen in the survey. I suggest the authors at least mention this as a week point of the study and the need of replicate these analyses in more colonies/locations.
No doubt you are right and a wider study of ant colonies will make the conclusions more reliable. We are planning this in the near future. The phrase about this has been added (Lines 488-490).
I am not an expert on the microbiology and biochemistry methods employed in this paper so I dont feel qualified to judge them. I hope that at least one other reviewer has adequate experience in those methods to offer constructive criticism. I hope
Some minor comments:
L53: I suggest replacing “is” with “are” and delete “taxon of”
The phrase was corrected (Line 60).
L60: Replace the first “the” with “a” and delete “ant”. Delete “of”
The phrases were corrected (Line 67).
L60-65: This paragraph contains information that should go in the abtract/results/discussion sections, not in the introduction.
The paragraph was reworked and moved in Discussion section (Lines 481-490).
L92: should be: The extraction of genomic DNA of isolates “and” PCR amplification were achieved…
The phrases were corrected (Line 110).
L205: delete “them”
The word was removed (Line 194).
L365: replace “has” with “have”
The typo was corrected (Line 418).
L398-399: Should be: The genus MessorForel, 1890 is a moderately large genus, “with” more than 126 species recognized, …
The phrases were corrected (Line 452).
